# Innate Immune Cells in the Tumor Microenvironment of Liver Metastasis from Colorectal Cancer: Contribution to a Comprehensive Therapy

**DOI:** 10.3390/cancers15123222

**Published:** 2023-06-16

**Authors:** Gabriela Sampaio-Ribeiro, Ana Ruivo, Ana Silva, Ana Lúcia Santos, Rui Caetano Oliveira, João Gama, Maria Augusta Cipriano, José Guilherme Tralhão, Artur Paiva

**Affiliations:** 1Flow Cytometry Unit, Clinical Pathology Department, Centro Hospitalar e Universitário de Coimbra EPE, 3000-075 Coimbra, Portugal; 2Institute for Clinical and Biomedical Research (iCBR), Faculty of Medicine, University of Coimbra, 3000-548 Coimbra, Portugal; 3Center for Innovative Biomedicine and Biotechnology (CIBB), University of Coimbra, 3000-548 Coimbra, Portugal; 4Surgery Department, Centro Hospitalar e Universitário de Coimbra, 3000-075 Coimbra, Portugal; 5Faculty of Medicine, University of Coimbra, 3000-548 Coimbra, Portugal; 6Germano de Sousa—Centro de Diagnóstico Histopatológico CEDAP, 3000-377 Coimbra, Portugal; 7Centre of Investigation on Genetics and Oncobiology (CIMAGO), Faculty of Medicine, University of Coimbra, 3000-548 Coimbra, Portugal; 8Clinical and Academic Center of Coimbra (CACC), 3000-075 Coimbra, Portugal; 9Pathology Department, Centro Hospitalar e Universitário de Coimbra, 3000-075 Coimbra, Portugal; 10Ciências Biomédicas Laboratoriais, ESTESC—Coimbra Health School, Instituto Politécnico de Coimbra, 3046-854 Coimbra, Portugal

**Keywords:** colorectal cancer, liver metastasis, innate immune system, monocytes, personalized medicine

## Abstract

**Simple Summary:**

In this study, the role of cells of the innate immune system in the development and progression of colorectal cancer liver metastasis was investigated. Two proteins, CD274 (PD-L1) and CD206 (MRC1), have been associated with poor prognosis and disease progression. Our study analyzed tumoral and non-tumoral biopsies from 47 patients with CRC liver metastasis using flow cytometry to phenotypically characterize innate immune cells. Comparing tumor with non-tumor samples, we found an increase in CD274 expression on classical, intermediate, and, in a lower extension, non-classical monocytes. Interestingly, we observed a decrease in the percentage of all monocyte subpopulations expressing CD274 and CD206 in tumor samples with a desmoplastic growth pattern compared to non-desmoplastic. Moreover, a lower percentage of monocyte subpopulations expressing CD206 or CD274 was associated with increased disease-free survival. Our study suggests potential new targets and biomarkers for precision medicine to improve CRC patients’ outcomes.

**Abstract:**

Colorectal cancer (CRC) is the third most prevalent type of cancer, and liver metastasis is the most common site of metastatic development. In the tumor microenvironment (TME), various innate immune cells are known to influence cancer progression and metastasis occurrence. CD274 (PD-L1) and CD206 (MRC1) are proteins that have been associated with poor prognosis and disease progression. We conducted a study on tumoral and non-tumoral biopsies from 47 patients with CRC liver metastasis, using flow cytometry to phenotypically characterize innate immune cells. Our findings showed an increase in the expression of CD274 on classical, intermediate, and non-classical monocytes when comparing tumor with non-tumor samples. Furthermore, tumor samples with a desmoplastic growth pattern exhibited a significantly decreased percentage of CD274- and CD206-positive cells in all monocyte populations compared to non-desmoplastic samples. We found a correlation between a lower expression of CD206 or CD274 on classical, intermediate, and non-classical monocytes and increased disease-free survival, which points to a better prognosis for these patients. In conclusion, our study has identified potential new targets and biomarkers that could be incorporated into a personalized medicine approach to enhance the outcome for colorectal cancer patients.

## 1. Introduction

The innate immune system is the first line of defense against invading pathogens [1]. This system is composed of various cells, tissues, and mechanisms that work together to identify and eliminate potential threats. The key components of the innate immune system include physical barriers such as skin and mucosal surfaces, cellular components such as phagocytes, and soluble factors such as cytokines and complements [2].

Colorectal cancer liver metastasis (CRCLM) refers to the spread of cancer from the colon or rectum to the liver. This is a common occurrence in patients with advanced colorectal cancer and can significantly impact their prognosis [3]. CRCLM is associated with a poorer prognosis than primary liver cancer and can be challenging to treat due to the complex liver microenvironment [4,5].

The liver microenvironment refers to the complex interplay of various cell types, extracellular matrix components, cytokines, and growth factors that exist within the liver. This microenvironment plays a critical role in the development, progression, and response to cancer therapy [6].

Several types of immune cells within the liver microenvironment can be characterized to identify new targets for creating new therapeutical approaches for patients with CRCLM.

Eosinophils, a type of white blood cells, are commonly present in high numbers in the microenvironment of CRCLM [7]. They play a significant role in the immune cells within the tumor microenvironment and have potential therapeutic and prognostic importance in human cancers [8]. Eosinophils have been detected in tumor-infiltrated regions and are associated with a positive prognosis, irrespective of factors such as stage, age, and histological grading [9,10].

Neutrophils, which are the most abundant type of white blood cells in the body, have been shown to promote cancer cell growth and invasion as well as angiogenesis in CRCLM [11]. It has been noted that these cells possess the ability to exhibit both pro-tumoral and anti-tumoral effects in response to various signals originating from tumors. However, the precise role of neutrophils in tumor progression still needs to be comprehended [9,12]. Elevated neutrophil levels are linked to a negative outlook in colorectal cancer patients, and targeting these cells is proposed as a therapeutic strategy for CRCLM treatment [13]. Neutrophils in the tumor microenvironment may serve as predictive markers for therapy response in CRCLM patients [14]. However, more research is needed to understand their role and develop effective targeting strategies.

Programmed death-ligand 1 (PD-L1), commonly referred to as CD274, is a protein located on the surface of specific cancer cells that helps control the immune system’s reaction to cancer [15]. High levels of PD-L1 expression have been linked to poor outcomes and resistance to therapy in several cancer types, such as gastric cancer and endometrial cancer [16,17]. Additionally, PD-L1 expression has been identified as a predictor of a patient’s response to immunotherapy for colorectal cancer [18]. As a result, PD-L1 inhibitors are currently being explored as a potential treatment option for this patient population through clinical development.

Furthermore, mannose receptor C type 1 (MRC1), also known as CD206, is a protein expressed on the surface of macrophages, a type of white blood cell that participates in the immune response to cancer [19]. Particularly, the M2 macrophage, with high CD206 expression, has been reported to promote tumor proliferation, angiogenesis, metastasis, and resistance to anti-cancer therapies [20,21]. Additionally, CD206 expression has been found to be a predictive marker for poor response to therapy and survival in patients with CRCLM [22,23]. Targeting CD206 has also been proposed as a potential strategy for treating CRCLM. Monocytes can migrate into tumors and differentiate into macrophages which can play a role in the progression of cancer and have been found to express CD206 [24,25]. Studies have shown that the presence of CD206-positive monocytes in the blood of colorectal cancer patients is associated with poor prognosis and poor response to therapy [26]. Therefore, targeting CD206 on monocytes has also been proposed as a potential strategy for treating CRCLM.

Moreover, a subpopulation of immune cells known as CCR6-positive cells has been linked to the development of CRCLM. Studies have also indicated that CCR6^+^ cells may help to create a favorable environment for the development and growth of cancer cells, also increasing angiogenesis and migration of cancer cells [27,28]. However, the exact mechanisms underlying the contribution of CCR6^+^ cells in CRCLM are still being studied and more research is needed to understand their role in this process.

Considering these variables, the objective of this study was the identification and phenotypic characterization of innate immune cells by flow cytometry in tissue samples obtained from patients with liver metastasis from colorectal cancer. These samples were further categorized as non-tumor and tumor samples based on their growth patterns, specifically non-desmoplastic or desmoplastic. Therefore, the main goal of our study was to gain a better understanding of the CRCLM tumor microenvironment, with an attempt to contribute to the identification of new potential targets for inhibiting disease progression and/or biomarkers for prognosis stratification.

## 2. Materials and Methods

### 2.1. Participants and Sample Collection

Non-tumor and tumor samples from 47 individuals with CRC liver metastasis were collected at the Centro Hospitalar e Universitário de Coimbra (CHUC). The demographic and clinical characteristics of the patients are detailed in Table 1. Prior to the study, the institution’s ethics committee approved the study (number CHUC-127-19), and all participants provided their signed consent after being informed of the study’s details. An experienced pathologist harvested fresh biological material within the first 30 min of surgical resection. The surgical specimen was meticulously sectioned, and approximately 3–5 mm^3^ of tumoral and non-tumoral tissue was collected from the tumor/liver interface. In addition, blood samples from 5 participants of the study were also collected on the day of the surgical resection and stained with the same protocol used for the tissue samples. The control blood samples were obtained from 5 healthy participants within the same range of age as the patients with CRCLM, who also gave their informed consent.

### 2.2. Histological Analysis

Concerning the analysis of liver metastasis samples from colorectal cancer (CRC), we assessed the growth pattern at the periphery of the liver metastasis, as described in our previous study [29]. We categorized the growth patterns as follows: desmoplastic growth pattern, characterized by the presence of connective tissue bordering the tumor and liver parenchyma; expansive growth pattern, where tumor compression causes deformation of liver cell plaques at the interface; and replacement growth pattern, where tumor cells infiltrate and replace hepatocytes within liver cell plaques. For a pattern to be considered, it needed to account for more than 75% of the interface. The growth pattern was further classified as either desmoplastic or non-desmoplastic, reflecting its biological behavior [30,31]. All slides were examined under a Nikon Eclipse 50i light microscope, and images were captured using a Nikon-Digital Sight DS-Fi1 camera (Figure 1a,b).

### 2.3. Flow Cytometry Characterization of Innate Immune Cells

#### 2.3.1. Staining Protocol

The non-tumor and tumor liver samples of CRC metastases were soaked in phosphate-buffered saline (PBS) (Gibco, Life Technologies in Paisley, UK) and repeatedly injected with PBS to release the maximum amount of cells into the supernatant. The suspensions were then collected in a 15 mL Falcon tube and centrifuged at 500× *g* for 5 min. The supernatant was discarded, and the pellet was resuspended in 1 mL of PBS. For the blood samples, we used a lyse and wash staining procedure.

To identify the various immune cell subpopulations of interest, the cell surface markers of each sample’s cellular suspension were stained using a direct immunofluorescence technique called “stain-lyse-wash”. An eight-color monoclonal antibody combination panel was performed for this purpose, and Table 2 provides details about the characteristics of the antibodies. The monoclonal antibodies were added to 300 μL of each sample (tumor and non-tumor cell suspension, as well as the blood samples) and incubated in the dark at room temperature for 10 min. Subsequently, 2 mL of FACSLysing solution (BD, Becton Dickinson Biosciences in San Jose, CA, USA) was added, and the samples were incubated in the dark for another 10 min at room temperature. After that, the samples were centrifuged at 500 *g* for 4 min. The supernatant was discarded, and the cell pellet was washed in 1 mL of PBS and resuspended in 500 μL of PBS.

#### 2.3.2. Sample Acquisition and Analysis

The samples were acquired on a FACSCanto II flow cytometer (BD), equipped with the FACSDiva software (v6.1.2; BD in San Jose, CA, USA). It is important to note that all results were obtained using the correct compensation controls, following the recommendations specified by the EuroFlow consortium [26]. For data analysis, the Infinicyt™ software (V.1.8; Cytognos SL Salamanca, Spain) was used. The gating strategy, to allow the identification of the various cell populations described in this study, is shown in Figure 2.

## 3. Results

### 3.1. Innate Immune Cells in Liver Colorectal Cancer Metastasis Microenvironment

To evaluate the immune microenvironment of CRCLM from both non-tumoral and tumor liver tissue samples, flow cytometry analysis was used, as described in Section 2. The distinct populations of innate immune cells were identified using specific markers associated with each population. A comparative analysis was performed between the non-tumor and tumor samples to assess discrepancies in the frequency of eosinophils, neutrophils, monocytes, lymphocytes, and NK cell populations under both conditions. In a supplementary analysis, the tumor samples were further divided into two groups based on the previously conducted histological characterization of the tumor growth pattern: tumor samples exhibiting a desmoplastic growth pattern and those with a non-desmoplastic pattern, which encompasses the expansive and replacement patterns.

According to our results, when we removed the non-hematopoietic cells from the total percentage of cells within each sample, the percentages of neutrophils, monocytes, and lymphocytes, which includes both B and T lymphocytes, do not present any significant differences in tumor samples compared to non-tumor samples (Table 3). Only eosinophils and NK cells showed a significantly decreased percentage in tumor samples compared to non-tumor samples (*p <* 0.0001), suggesting the presence of lower numbers of immune cells with a cytolytic activity that could favor the colorectal cancer cells’ colonization and proliferation in the liver.

No significant differences were noted when we compared tumor samples with desmoplastic and non-desmoplastic growth patterns.

Moreover, the expression of CD274 and CD206 was evaluated on the different monocyte subpopulations: classical, intermediate, and non-classical monocytes. When comparing tumor samples with non-tumor samples, we did not observe significant differences in the percentage of monocyte subpopulations expressing CD274 or CD206 (Table 3).

However, we observed a significant decrease in the percentage of CD206^+^ classical and intermediate monocytes in tumor samples with a desmoplastic growth pattern compared to samples with non-desmoplastic ones (*p* < 0.05). The percentage of CD274^+^ classical, intermediate, and non-classical monocytes was also significantly decreased in tumor samples with a desmoplastic growth pattern (*p* < 0.05).

### 3.2. Mean Fluorescence Intensity (MFI) Analysis of CD206 and CD274 Reveals Differential Expression on Cells from CRC Liver Metastasis

An increased CD274 expression on classical and intermediate monocytes from tumor samples compared to non-tumor samples (*p* < 0.01 and *p* < 0.05, respectively) was observed. The same tendency was also found for CD206 expression on classical monocytes from tumor samples compared to non-tumor samples, however not reaching statistical significance (Figure 3a–c).

Additionally, our results also showed that tumor samples with a desmoplastic growth pattern have a lower expression of CD206 and CD274 compared to tumor samples with non-desmoplastic tumor growth, being significantly different for CD274 expression on non-classical monocytes (Figure 3d–f).

The expression of CD274 and CCR6 was also evaluated on non-hematopoietic cells present in both non-tumor and tumor samples, as well as in tumor samples with non-desmoplastic or desmoplastic growth patterns (Figure 3g–j). A decreased expression of CD274 and CCR6 (*p* < 0.001) on non-hematopoietic cells from tumor samples when compared with non-tumor samples was observed.

Comparing tumor samples with a desmoplastic growth pattern with non-desmoplastic samples, an increased MFI of CD274 on non-hematopoietic cells within desmoplastic growth pattern tumor samples from CRCLM patients was observed (*p* < 0.01). Our results also reveal a tendency to an increase in CCR6 expression on non-hematopoietic cells from desmoplastic samples.

### 3.3. Blood Sample Analysis of CRCLM Patients

In the attempt to understand if the expression of CD206 and CD274 in all monocyte subpopulations increases from the ones that circulate in peripheral blood to those located in non-tumor and tumor microenvironments, we analyzed a small cohort of five peripheral blood samples from CRCLM patients and five matched controls (paired in terms of age and sex distribution).

The results show no significant differences in the percentage of total monocytes, as well as in the relative frequencies of each monocyte subpopulation, in blood samples from CRCLM patients compared to control blood samples (Figure 4a–d).

Furthermore, the percentage of all monocyte subpopulations expressing CD206 and CD274, as well as the expression of both markers, were also evaluated.

This analysis showed an overall decrease in the percentage of monocyte subpopulations expressing CD206 or CD274, as well as in the expression of these markers (MFI) in the blood samples, either from CRCLM or control (CTR), when compared to the tissue samples (non-tumor or tumor) from CRCLM patients (Figure 5a–l). These results seem to indicate that the differentiation of monocytes to M2 macrophages occurs in the tissue, particularly at the tumor side.

### 3.4. The Percentage of CD206^+^ and CD274^+^ Monocytes Influences the Disease-Free Survival Time of CRCLM Patients

Considering the results above, we noticed an interesting feature in both non-tumor and tumor samples from the CRCLM patients. The analysis revealed the establishment of two groups based on the percentage of classical, intermediate, and non-classical monocytes expressing CD206 or CD274: a group of patients with a higher percentage of CD206^+^ or CD274^+^ and another group with a lower percentage of each marker.

After carefully analyzing these results and establishing the different groups of patients based on more or less than 70% of monocytes expressing CD206 and more or less than 50% of monocytes expressing CD274, we found that the overall disease-free survival of these patients are significantly different (Figure 6). The group of patients with a lower percentage of these markers showed an increased time of disease-free survival compared to the patients with higher percentages of monocytes expressing these markers (*p* < 0.05).

This information could have significant clinical implications as it may help to predict the course of the disease and develop more effective treatment strategies for these patients.

## 4. Discussion

Colorectal cancer (CRC) is gaining increasing global attention as it has emerged as the third most prevalent cancer and the second most fatal [32], in which liver metastatic formation is the main cause of death [33]. Despite numerous studies focusing on the mechanism of CRCLM, the specific underlying mechanism remains unclear. Elucidating this mechanism could significantly contribute to the prevention and treatment of CRCLM.

Immune cells in the tumor microenvironment play a role in the advancement and spreading of cancer cells. The innate immune system is especially significant in the development of colorectal cancer liver metastasis [6,11,34].

This study extensively analyzed and identified diverse populations present in human samples of liver metastasis from CRC. To achieve this, flow cytometry analysis was used due to its ability to handle multiple parameters effectively and enhance the reproducibility of results.

Studies have suggested that a higher percentage of eosinophils within the tumor microenvironment may be associated with a better prognosis in CRCLM patients [7,35]. This is thought to be due to the ability of eosinophils to attack and destroy cancer cells as well as to stimulate the immune response against the tumor [36]. On the other hand, an increased number of neutrophils within the tumor microenvironment may be associated with a poorer prognosis in CRCLM patients [14,37]. Neutrophils release factors that promote tumor growth and progression, being able to suppress the immune response against the tumor [38]. Our findings revealed that tumor samples from CRCLM patients have increased neutrophil percentage compared to the eosinophil population, which indicates that the eosinophils/neutrophils ratio has decreased, indicating a poor prognosis for these CRCLM patients. In addition, the tumor samples presented a significantly decreased percentage of eosinophils compared to non-tumor samples, reinforcing poor outcomes for CRCLM patients. When we compared tumor samples with different growth patterns, no significant differences were observed between the two groups. These results may indicate that the balance between eosinophils and neutrophils within the tumor microenvironment can be a critical factor in determining the prognosis of CRCLM patients, but not to differentiate the patients’ outcome relative to their tumor growth pattern.

The presence of lymphocytes is also a parameter to take into account for the evaluation of the progression of CRC cells. Therefore, an increased percentage of lymphocytes has been linked to a better prognosis [39,40]. Our results showed no significant differences in the percentage of these cells (T and B cells) between non-tumor and tumor samples. In addition to this, the percentage of NK cells was also evaluated. NK cells are a type of white blood cell that plays a crucial role in the immune response against cancer cells [41,42]. Their function is to recognize and destroy abnormal or infected cells, including cancer cells, through various mechanisms. They release cytotoxic granules that can induce apoptosis in target cells, and secrete cytokines that activate other immune cells and enhance the immune response against cancer [43]. The presence of these cells is significantly decreased in the tumor samples of CRCLM, which contributes to an immunosuppressive microenvironment creation that favors CRC cells’ progression, proliferation, and establishment in the liver.

Additionally, as a type of white blood cell that plays an important role in the immune system, monocytes are known to infiltrate tumors and contribute to tumor growth and progression by promoting angiogenesis, immune suppression, and tissue remodeling [44,45,46,47].

Several studies have investigated the relationship between monocytes and cancer progression, and some studies have suggested that monocytes/macrophages could be trained by tumor cells to become metastasis-promoting by inducing the epithelial–mesenchymal transition (EMT), extracellular matrix remodeling, and cancer cell intravasation/extravasation [48,49,50,51]. Other studies have also found that monocytes can contribute to the metastasis of cancer cells by forming a pre-metastatic niche in distant organs [48,52,53], but further research is needed to fully understand the mechanisms involved and to identify potential therapeutic targets. Overall, these studies suggest that monocytes can play a significant role in the development of metastasis in cancer patients by promoting inflammation and stimulating the growth and survival of cancer cells. In this study, we revealed that the percentage of monocytes within the tumor samples of CRCLM patients has no significant differences when compared to the non-tumor samples, and also when we compared the monocyte percentage in both tumor samples with desmoplastic and non-desmoplastic growth patterns.

Interestingly, the percentage of CD206- and CD274-positive monocytes showed no differences between non-tumor and tumor samples of CRCLM patients. However, when we evaluated the expression of these markers on non-desmoplastic and desmoplastic growth patterns, the analysis revealed a significantly decreased expression in tumor samples with desmoplastic growth patterns.

The expression of CD274 (also known as PD-L1) on monocytes has been studied concerning prognosis in patients with colorectal cancer [54,55]. CD274 is a protein that is involved in immune regulation and can be expressed on the surface of immune cells, including monocytes [55,56]. Several studies have reported that high levels of CD274 expression on monocytes are associated with poor prognosis in patients with colorectal cancer [54]. In contrast, other studies have reported that the expression of CD274 was an independent factor for the patient’s prognosis [57,58], which demonstrates that the expression patterns of immune checkpoints in macrophages and their clinical value are poorly established in colorectal cancer and further studies are needed for targeting them.

CD206 (also known as MRC1) is a scavenger receptor that is primarily expressed on the surface of macrophages, including monocytes that differentiate into macrophages [59,60]. CD206 is involved in the clearance of cellular debris and immune response regulation. Studies have investigated the expression of CD206 on monocytes concerning prognosis in patients with colorectal cancer, revealing that CD206 expression on monocytes was associated with tumor progression in patients with colorectal cancer, the CD206 expression being significantly higher in patients with advanced-stage tumors compared to those with early-stage tumors [61,62].

Moreover, these two markers have been associated with an M2-like macrophage population that indicates the promotion of an immunosuppressive and pro-tumor phenotype in colorectal cancer [63,64]. Taking this into consideration, we can infer that CRCLM patients with tumors with desmoplastic growth patterns are more likely to have a better prognosis in the progression of the disease, suggesting that the histological analysis of the tumor samples is crucial to understand the progression of the CRCLM and possibly the strategy of treatment used for each case.

In this study, the expression of CD274 and CD206 was also evaluated. The levels of expression of CD274 were significantly increased on classical and intermediate monocytes, and, in a lower extension, on non-classical monocytes.

Specifically, the PD-1/PD-L1 signaling pathway activation may be the mechanism by which tumors escape the antigen-specific T-cell immune response. By interacting with PD-L1 on tumor cells, PD-1 expressed on immune cells can protect tumor cells, preventing their elimination by the immune cells [65]. Additionally, some cancer cells and other abnormal cells can overexpress PD-L1, which can suppress the immune response against them and allow them to evade the immune system. This is why drugs that target the PD-1/PD-L1 pathway, known as immune checkpoint inhibitors, have been developed and are used to treat certain types of cancer [66]. In addition to its role in regulating the immune system, PD-L1 has also been shown to be involved in other processes, such as tissue inflammation and autoimmune responses [67]. PD-L1 expression levels in tumors have been used as a predictive biomarker for response to immune checkpoint inhibitors in several cancer types, including melanoma, non-small cell lung cancer, bladder cancer, and colorectal cancer [68,69,70].

Accordingly, this study revealed that non-hematopoietic cells presented significantly decreased CD274 expression on tumor samples from CRCLM patients compared to non-tumor samples. However, depending on the tumor sample growth pattern, the expression of CD274 changes. In desmoplastic tumor samples, the expression of CD274 is significantly increased compared to non-desmoplastic tumor samples, suggesting that CRCLM patients with desmoplastic growth patterns should benefit from therapy with an immune checkpoint inhibitor for PD-L1.

Moreover, the expression of both CD206 and CD274 is significantly decreased in blood samples of CRCLM patients compared to tissue samples, and no significant differences were noted between the CRCLM patients’ blood samples and the control ones, indicating that the differentiation of monocytes to M2 macrophages mostly occurs in the tissues, particularly at the tumor site [63,71].

After a deeper evaluation of the results, we verified that patients with lower percentages of monocytes expressing CD206 or CD274 exhibited increased overall disease-free survival rates when compared to patients with higher percentages of monocytes expressing these markers. This information could have significant clinical implications as it may help to predict the course of the disease and could allow the introduction of more effective treatment strategies for these patients. It may also help to identify patients who are more likely to benefit from certain therapies, such as immunotherapies or targeted therapies, that can boost the immune response against cancer cells.

Overall, our study results are in agreement with the existing literature, where emerging evidence indicates that the presence of CD206- and CD274-positive monocytes in the liver metastatic site of CRC patients correlates with aggressive disease progression and unfavorable clinical outcomes. Zhang et al. conducted a flow cytometry analysis and identified a CD206^+^ macrophage population, with approximately 70% of these cells present in tumor samples of CRC patients [72]. Similar findings were observed in other types of cancers such as lung, breast, ovarian, and prostate cancer [73,74,75].

In addition, genomic approaches have been made to establish this relationship between the expression of these two markers in monocytes and tumor-infiltrating macrophages and the poor prognosis of CRCLM patients. For example, Yin et al. showed that a specific PD-L1^+^ CD206^+^ macrophage subgroup induced by CRC-derived extracellular vesicles-miRNAs could predict a poor prognosis in CRC [63]. This tumor-associated macrophage (TAM) subgroup promotes tumor growth by inhibiting CD8^+^ T cell activity and thus inducing an immunosuppressive microenvironment. In addition, Geng et al. also identified TAMs with increased expression of CD206 as a potential immunotherapeutic target for CRCLM patients [76]. This kind of study and several others that characterize the tumor immune microenvironment with genomic and transcriptomic approaches may serve as novel methods for CRC treatment and specifically in liver metastasis inhibition [77,78,79,80,81].

The chemokine receptor CCR6 is believed to play a role in promoting CRCLM [27,82]. Studies suggest that CCR6 may facilitate the interaction between CRC cells and liver stromal cells, leading to the formation of a pro-metastatic microenvironment and promoting the migration and invasion of CRC cells [83,84]. CCR6 has been proposed as a potential strategy for inhibiting CRC liver metastasis. Even while CCR6 is not considered a typical checkpoint inhibitor, some evidence suggests that targeting CCR6 may enhance the efficacy of immune checkpoint inhibitors in treating certain types of cancer, including CRC [85]. The expression of CCR6 is decreased in non-hematopoietic cells present in tumor samples of CRC liver metastasis patients compared to non-tumor samples, but elevated in tumor samples with a desmoplastic growth pattern compared to non-desmoplastic samples. Targeting CCR6 on tumor cells may be a potential strategy for inhibiting CRC metastasis by enhancing the efficacy of immune checkpoint inhibitors, but further research is needed to fully understand the role of CCR6 in CRC and its metastasis.

This study has some limitations. Although developed in a reference center for oncologic liver surgery with a high volume of cases per year, it is a single-center study with its inherent lack of external validation. Additionally, the sample size of the blood samples’ group is small, which limits the ability to draw definitive conclusions. One of the limitations of histologic analysis is that as good as the conclusions might be, they can only be reached through specimen analysis. The identification of liver metastasis histopathological growth patterns, which have demonstrated prognostic significance in previous studies, requires tissue sampling through liver resection or other invasive methods. As an alternative approach, liquid biopsy has emerged as a promising method for obtaining biomarkers and has been widely employed in cancer management [86,87,88]. Given this, one of the objectives of our preliminary blood analysis was to understand if the frequencies of monocyte subpopulasstions were different in CCRLM patients since increased numbers of non-classical monocytes have been associated with inflammation and tissue/organ injury. Moreover, it was an attempt to understand if the expression of CD274 and CD206 on monocytes increased from peripheral blood to TME, or even if they could constitute good peripheral biomarkers of disease progression. Another limitation of this study is the relatively short follow-up period, which can underestimate the conclusions. Prolonged follow-up of the participants is necessary and may add significant scientific value. In addition to these limitations, this study’s results reflect the complexity of immune cell interaction within the tumor microenvironment. Therefore, the interpretation of the results is not equally simple. Further studies are needed for a better understanding of causes and consequences in all of this complex process.

In terms of clinical implications, the results of this work represent another step towards a better understanding of CRC liver metastasis physiopathology and personalized medicine. These results are one step closer to impacting the therapeutic approach for these patients. The knowledge of the tumor microenvironment may allow us to modulate inflammatory responses and direct therapy for CRC liver metastases, with better results and improved survival. The pre-operative knowledge of the histological growth pattern would allow us to better select patients for surgery, since we know that desmoplastic growth pattern is associated with increased survival, and to supplement our strategy with the parenchymal sparing option in the desmoplastic growth pattern patients and more aggressive surgery with larger surgical margins in the non-desmoplastic. The extensive cellular characterization of the CRC liver metastases microenvironment reported here will contribute to precision medicine in the near future.

## 5. Conclusions

In summary, this study revealed that the tumor microenvironment, particularly the innate immune system, seemed to play a crucial role in the progression of CRCLM. Overall, we found that the tumor microenvironment of CRCLM has a significantly decreased percentage of eosinophils and NK cells, which could contribute to the development of an immunosuppressive microenvironment.

We identify an altered expression of CD274 and CD206 in classical, intermediate, and non-classical monocytes between tumor and non-tumor samples, as well as between desmoplastic and non-desmoplastic growth patterns, indicating that patients with a desmoplastic growth pattern are more likely to have a better prognosis and benefit from anti-PD-L1 immunotherapy.

Overall, this study highlights the importance of a deep characterization of the tumor microenvironment by understanding the composition of different leukocyte populations, which could contribute to a comprehensive therapy, particularly in this new era of precision medicine, and establishing potential targets for new therapeutic approaches.

## Figures and Tables

**Figure 1 cancers-15-03222-f001:**
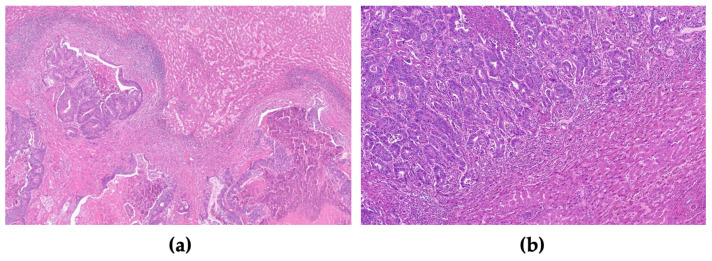
Liver gross section with colorectal metastases and representative histological images of the growth pattern of liver metastasis of CRC. (**a**) Representative images of the immunohistochemistry-stained tumor sections with desmoplastic growth pattern (H&E 20×) and (**b**) non-desmoplastic growth pattern (H&E 100×).

**Figure 2 cancers-15-03222-f002:**
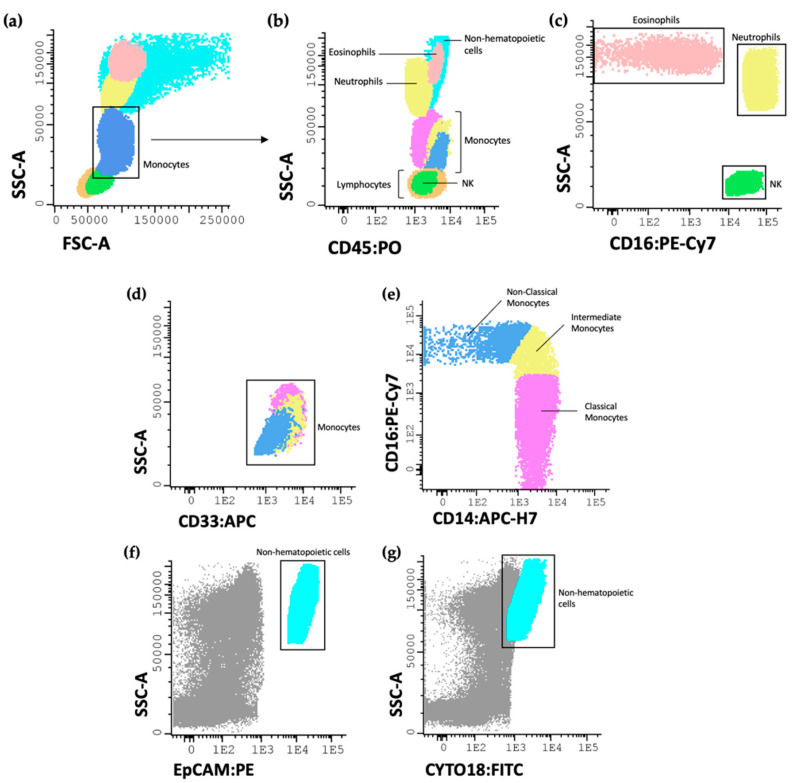
Representative dot plot histograms showing the gating strategy used for the identification of the studied immune cells. Eosinophils, neutrophils, monocytes, and lymphocytes were first identified by their size and complexity (**a**) and by CD45 (**b**); the eosinophil population was also identified due to the absence of CD16 expression; the neutrophil and NK cell populations were identified based on the bright expression of CD16 (**c**); the monocyte populations were first assessed by their bright expression of CD33 (**d**) and their subpopulations identified as classical (CD14^+^ CD16^−^), intermediate (CD14^+^ CD16^+^) and non-classical (CD14^−^ CD16^++^) (**e**); non-hematopoietic cells were identified as EpCAM^+^ (**f**) and CYTO18^+^ (**g**). EpCAM or CD326 (epithelial cell adhesion molecule); CYTO 18 (cytokeratin 18).

**Figure 3 cancers-15-03222-f003:**
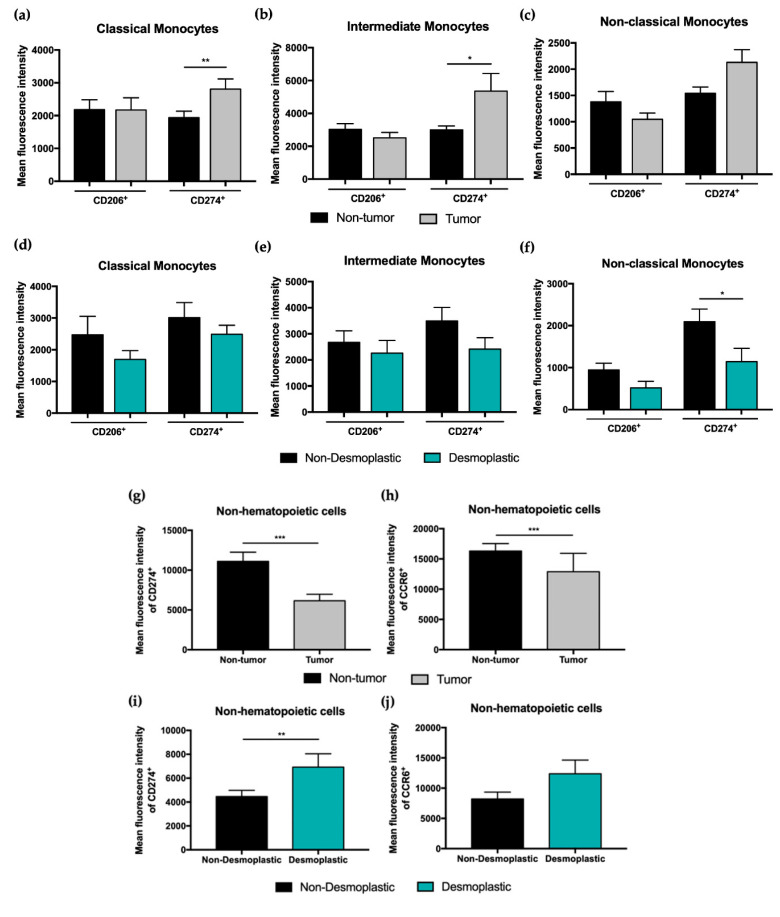
Flow cytometry analysis of the mean fluorescence intensity of different cell populations present in CRC liver metastasis patients’ samples. Mean fluorescence intensity of CD206^+^, CD274^+^, and CD33^+^ in classical (**a**), intermediate (**b**), and non-classical monocytes (**c**) in non-tumor and tumor samples from CRC liver metastasis patients. Mean fluorescence intensity of CD206^+^, CD274^+^ and CD33^+^ in classical (**d**), intermediate (**e**), and non-classical monocytes (**f**) in tumor samples with a desmoplastic or non-desmoplastic growth pattern. Analysis of the mean of fluorescence intensity of CD274^+^ and CCR6^+^ in non-hematopoietic cells present in non-tumor and tumor samples from CRC liver metastasis patients (**g**,**h**), and in tumor samples with a desmoplastic or non-desmoplastic growth pattern (**i**,**j**). All results are shown as mean ± SEM. * *p* < 0.05, ** *p* < 0.01, and *** *p* < 0.001 were significantly different when compared to non-tumor samples or non-desmoplastic tumor samples using Mann–Whitney tests.

**Figure 4 cancers-15-03222-f004:**
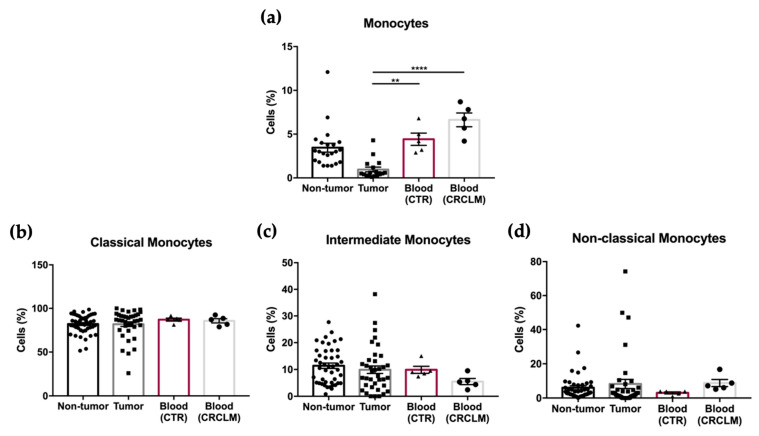
Flow cytometry analysis of the percentage of total monocytes (**a**), and classical (**b**), intermediate (**c**), and non-classical monocytes (**d**) in non-tumor and tumor samples, compared to control blood samples (CTR) and blood samples from CRCLM patients (CRCLM). All results are shown as mean ± SEM. ** *p* < 0.01, and **** *p* < 0.0001 were significantly different when the blood samples were compared to the tumor samples from CRCLM patients using a one-way ANOVA followed by a non-parametric Kruskal-Wallis test for multiple comparisons.

**Figure 5 cancers-15-03222-f005:**
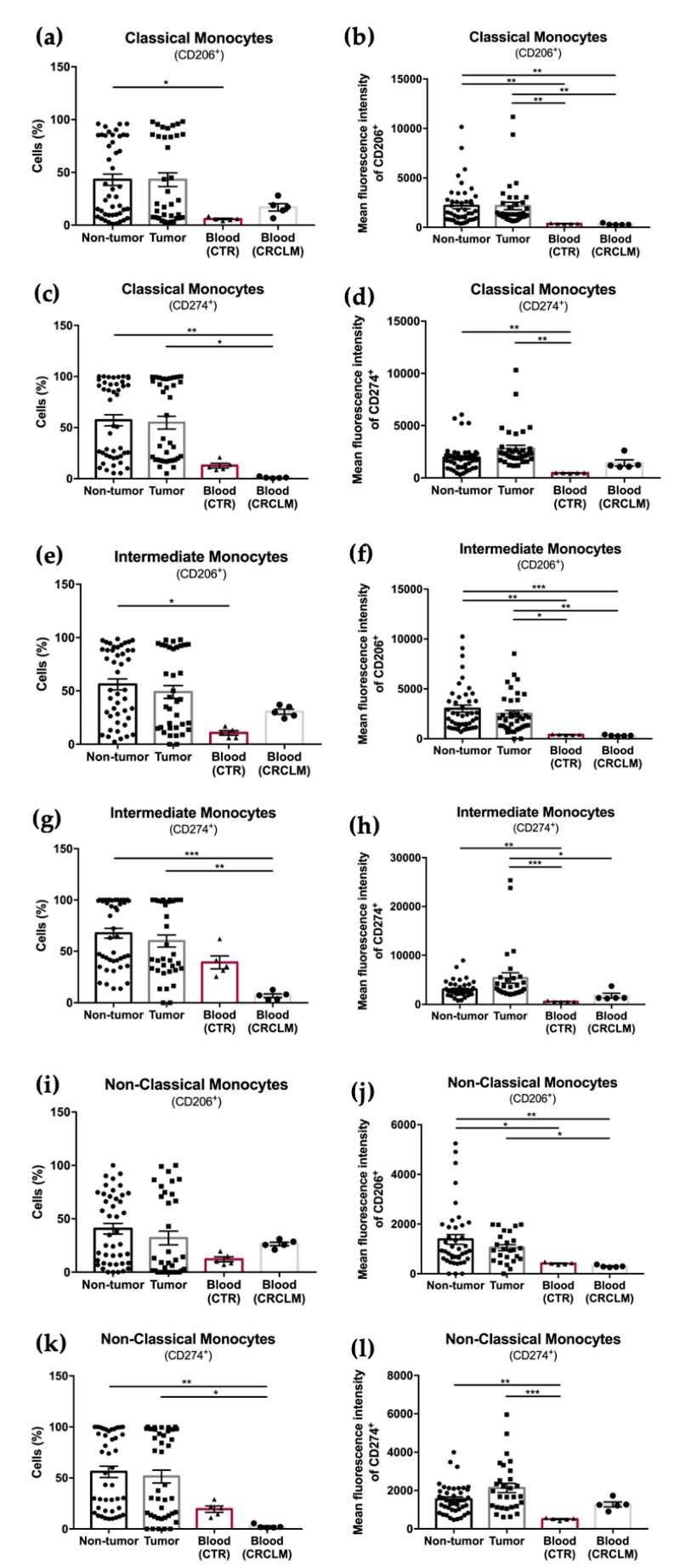
Flow cytometry analysis of the percentage and mean fluorescence intensity of different monocyte subpopulations present in non-tumor and tumor samples from CRCLM patients and in control and CRCLM patients’ blood samples. Percentage of CD206^+^ classical monocytes (**a**) and respective MFI (**b**); percentage of CD274^+^ classical monocytes (**c**) and MFI (**d**). Percentage of CD206^+^ intermediate monocytes (**e**) and respective MFI (**f**); percentage of CD274^+^ intermediate monocytes (**g**) and MFI (**h**). Percentage of CD206^+^ non-classical monocytes (**i**) and respective MFI (**j**); percentage of CD274^+^ non-classical monocytes (**k**) and MFI (**l**). * *p* < 0.05, ** *p* < 0.01, and *** *p* < 0.001 were significantly different when compared to non-tumor or tumor samples using Mann–Whitney tests.

**Figure 6 cancers-15-03222-f006:**
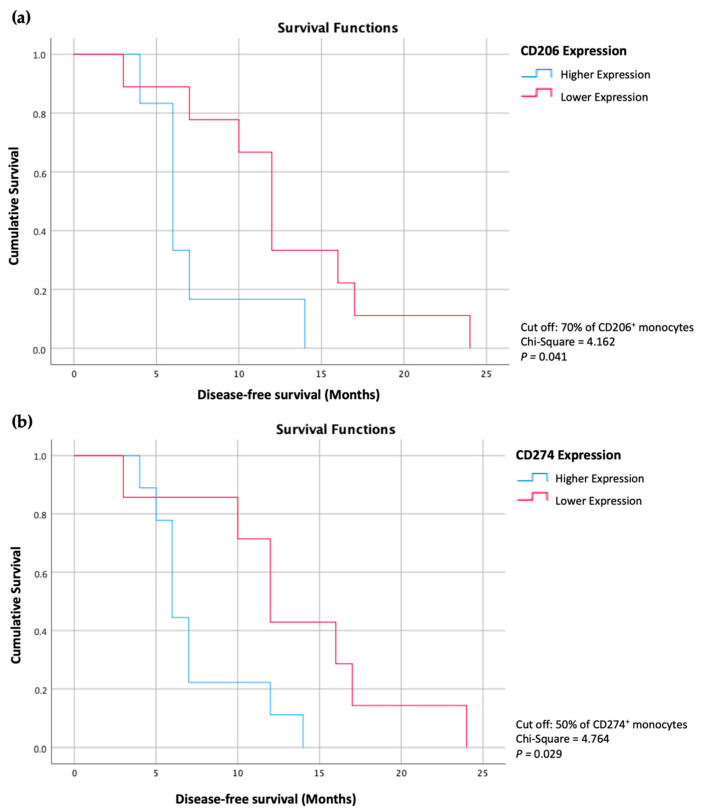
Kaplan-Meier curves representing the disease-free survival of CRCLM patients with a higher percentage of CD206^+^ monocytes compared with patients with a lower percentage of CD206^+^ monocytes (**a**) and the disease-free survival of CRCLM patients with a higher percentage of CD274^+^ monocytes compared with patients with a lower percentage of CD274^+^ monocytes (**b**).

**Table 1 cancers-15-03222-t001:** Demographic and clinical characteristics of the patients of the study.

Variable	(*n* = 47)
**Age at surgical resection**	
Mean ± SD; range	59.1 ± 12.8; 36–90
**Variable**	**Number (%)**
**Gender**	
Male	31 (66)
Female	16 (34)
**Presentation**	
Synchronous	26 (55)
Metachronous	21 (45)
**Primary tumor location**	
Sigmoid	16 (34)
Ascending	10 (21)
Descending	5 (11)
Splenic Flexure	4 (9)
Hepatic Flexure	1 (2)
Transverse	2 (4)
Rectal	9 (19)
**T stage of the primary colorectal tumor**	
3	33 (70)
4a	12 (26)
4b	2 (4)
**First approach**	
Liver First	6 (13)
Colectomy	34 (72)
Synchronous resection	7 (15)
**Preoperative systemic chemotherapy**	27 (57)
**Variable**	**(*n* = 47)**
**Number of colorectal liver metastases**	
Mean ± SD; range	2.3 ± 1.8; 1–10
**Size of the largest colorectal liver metastases (cm)**	
Mean ± SD; range	3.8 ± 2.4; 0.4–12
**Variable**	**Number (%)**
**Histologic growth pattern**	
Desmoplastic	17 (36)
Non-desmoplastic	30 (64)
**Status**	
Dead	3 (6)
Alive	44 (94)

**Table 2 cancers-15-03222-t002:** The panel of monoclonal antibodies used for the innate immune cells’ characterization, indicating their respective volume, the fluorochrome, commercial source, and clone.

Fluorochromes								
Tube	PB	PO	FITC	PE	PerCP-Cy5.5	PE-Cy7	APC	APC-H7
1	CD274 Biolegend (29E.2A3) (5 µL)	CD45 BD Biosciences (2D1) (2.5 µL)	Cito18 Cytognos (Ks18.04) (2.5 µL)	EpCAM BD Biosciences (EBA-1) (10 µL)	CD206 Biolegend (15–2) (5 µL)	CD16 BD Pharmingen (3G8) (2 µL)	CD33 BD Biosciences (P67.6) (2.5 µL)	CD14 BD Biosciences (mφp9) (2.5 µL)

Abbreviations: APC, allophycocyanin; APC-H7, allophycocyanin-hilite 7; FITC, fluorescein isothiocyanate; PB, pacific blue; PE, phycoerythrin; PE-Cy7, phycoerythrin-cyanine 7; PerCP-Cy5.5, peridinin chlorophyll protein-cyanine 5.5; PO, pacific orange. Commercial sources: BD Biosciences (Becton Dickinson Biosciences, San Jose, CA, USA); BD Pharmingen (San Diego, CA, USA); Biolegend (San Diego, CA, USA); Cytognos (SL Salamanca, Spain).

**Table 3 cancers-15-03222-t003:** Percentage of immune cells in non-tumor and tumor samples of liver metastasis from CRC.

Cell Types	Non-Tumor (%) ± SEM	Tumor (%) ± SEM	*p* Value	Tumor		*p* Value
				Non-Desmoplastic (%) ± SEM	Desmoplastic (%) ± SEM	
**Eosinophils**	0.53 ± 0.17	0.01 ± 0.01	0.0001 ^a^	0.002 ± 0.002	0.02 ± 0.01	0.38
**Eosinophils** (after removing non-hematopoietic cells)	0.81 ± 0.26	0.04 ± 0.03	0.0001 ^a^	0.01 ± 0.01	0.11 ± 0.08	0.38
**Neutrophils**	22.50 ± 2.50	6.87 ± 1.56	<0.0001 ^b^	5.63 ± 1.57	4.20 ± 1.59	0.38
**Neutrophils** (after removing non-hematopoietic cells)	34.88 ± 3.88	32.66 ± 7.43	0.50	27.65 ± 7.71	24.03 ± 9.05	0.64
**Lymphocytes**	31.36 ± 2.93	11.87 ± 1.99	<0.0001 ^b^	12.17 ± 1.62	11.96 ± 4.30	0.44
**Lymphocytes** (after removing non-hematopoietic cells)	48.61 ± 4.53	56.44 ± 9.44	0.79	59.75 ± 7.94	68.44 ± 24.62	0.67
**Monocytes**	3.45 ± 0.50	0.96 ± 0.27	<0.0001 ^b^	0.88 ± 0.29	0.55 ± 0.16	0.77
**Monocytes** (after removing non-hematopoietic cells)	5.35 ± 0.78	4.56 ± 1.27	0.07	4.33 ± 1.44	3.17 ± 0.93	0.93
Classical	82.08 ± 1.63	81.87 ± 2.68	0.43	80.37 ± 3.97	84.07 ± 3.19	0.96
CD206^+^	43.04 ± 5.33	43.04 ± 6.46	0.93	49.70 ± 7.99	22.16 ± 9.36	0.02 *
CD274^+^	57.03 ± 5.50	54.86 ± 6.26	0.82	62.40 ± 7.52	43.01 ± 10.49	0.03 *
Intermediate	11.38 ± 1.00	9.91 ± 1.40	0.17	10.85 ± 2.18	8.54 ± 1.33	0.98
CD206^+^	56.06 ± 5.08	48.95 ± 6.01	0.50	58.94 ± 7.19	28.29 ± 8.57	0.02 *
CD274^+^	67.53 ± 4.80	59.93 ± 5.90	0.25	66.14 ± 7.47	41.96 ± 8.80	0.04 *
Non-Classical	6.01 ± 1.16	8.22 ± 2.66	0.17	8.81 ± 3.84	7.35 ± 3.50	0.38
CD206^+^	40.71 ± 5.00	31.97 ± 6.44	0.09	34.37 ± 7.91	22.86 ± 10.50	0.23
CD274^+^	56.04 ± 5.54	51.42 ± 6.32	0.46	54.60 ± 9.09	29.72 ± 10.73	0.03 *
**NK cells**	4.25 ± 0.46	0.37 ± 0.06	<0.0001 ^b^	0.36 ± 0.08	0.29 ± 0.07	>0.99
**NK cells** (after removing non-hematopoietic cells)	6.60 ± 0.71	1.74 ± 0.29	<0.0001 ^b^	1.75 ± 0.39	1.65 ± 0.38	0.44
**Non-Hematopoietic cells**	35.49 ± 3.63	78.97 ± 2.37	<0.0001 ^b^	79.64 ± 2.85	82.52 ± 4.19	0.72

^a^ *p* < 0.001 and ^b^
*p* < 0.0001: significantly different when compared to non-tumor samples, using Mann–Whitney non-parametric tests; * *p* < 0.05: significantly different when compared to tumor samples with a non-desmoplastic growth pattern also using Mann–Whitney non-parametric tests. All results are shown as mean ± SEM.

## Data Availability

The data presented in this study are available in this article.

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
