# Peer review of "Innate Immune Cells in the Tumor Microenvironment of Liver Metastasis from Colorectal Cancer: Contribution to a Comprehensive Therapy"

_cancers, 2023, doi:10.3390/cancers15123222_

Round 1

Reviewer 1 Report

The whole manuscript should be edited for english-language usage (professional-editing)

The introduction should be shortened and focused on the main topic

A proper checklist must be added (see equatornetwork)

The discussion is extremely repetitive (especially on the definition of CRC). It should be shortened by briefly summarizing the study findings and comparing them with literature data

The limitations of the study should be added

A small paragraph on the clinical implications needs to be added

The references must be updated by adding the following articles and many others: 

The Role of Tumor Microenvironment Cells in Colorectal Cancer (CRC) Cachexia. Int J Mol Sci. 2021 Feb 4;22(4):1565. doi: 10.3390/ijms22041565

Therapeutic Targets and Tumor Microenvironment in Colorectal Cancer. J Clin Med. 2021 May 25;10(11):2295. doi: 10.3390/jcm10112295. PMID: 34070480; PMCID: PMC8197564.

Extensive editing

Reviewer 2 Report

The authors have prepared a well-organized, well-done study in MS form.  I have no issues with the data that are presented or the conclusions that are drawn (with the exception of the too small n=5 used for the blood study).

My main issues are that the MS is entirely descriptive, but more importantly the MS does not break much new ground.  Cancers is a high impact, high profile journal, and this paper should offer new, significant findings in the field of TME of CRC metastases.  The authors should be encouraged to build upon their findings to make a more novel, impactful study in this regard.

Reviewer 3 Report

The manuscript "Innate immune cells in the tumor microenvironment of liver metastasis from colorectal cancer: contribution to a comprehensive therapy" aims to study the innate immune cell populations within the tumor microenvironment in the case of colorectal cancer liver metastasis. The ultimate goal is to identify novel targets for developing treatment strategies. Sampaio-Ribeiro et al. use flow cytometry data to characterize the cell populations, and do a comparative analysis of non-tumor, tumor, and desmoplastic and non-desmoplastic tumor samples.

Overall, the manuscript is well written with a good study design. However, I strongly feel there is lot of room for improvement and I suggest some ways in the major and minor comments below.

Major comments:

1.     The authors have performed a lot of qualitative comparisons with the results and existing literature. How does the study results (wherever applicable) compare quantitatively with the previous studies in CRCLM? This is important to discuss the reasons for any differences. Also, a qualitative discussion on how the results compare with other tumors would be a good addition. There are some references in the Discussion section pointing to this, but a summary discussion would be helpful. Particularly because this manuscript is aiming for a comprehensive therapy.

2.     Did the authors try to search existing genomics data on CRC/CRCLM to support some of their results? Of course, a detailed bioinformatics/genomics analyses would be a whole new manuscript, but a simple survey of processed sequencing data like RNA-seq expression values and adding comments based on that would be a good head start for future genomics work. TCGA database or other independent studies have such data available.

Minor comments:

1.     Couple of numbers in Table 3 have “comma” for decimal notation. NK cells Tumor value and Classical P value. There may be more in the manuscript, the authors need to check.

2.     Please refer to Materials and Methods sections wherever appropriate throughout the Results section.

3.     Line 308: change to “These results clearly…”

4.     Line 328: remove second ‘based’ in “…based on more or less than based 70%...”

5.     Section 3.4 is not referring to Figure 6 anywhere in the text.

6.     There is a refresh button picture near x-axis label of Fig. 6b?

Just few minor edits as mentioned in the "Minor comments" above.

Round 2

Reviewer 1 Report

I'm satisfied with the changes made

Moderate

Reviewer 2 Report

The MS is well written, and while still descriptive in nature, it does address very timely and important issues in this field.

Reviewer 3 Report

The comments have been addressed.